

# Hematology and clinical biochemistry reference intervals for companion pigs using the ADVIA 2120 and Cobas c501

Deanna M. W. Schaefer[1], Ricardo Videla[2], Joe S. Smith[2],
Pierre-Yves Mulon[2], Bente Flatland[1] and Xiaojuan Zhu[3]

[1] Department of Biomedical and Diagnostic Sciences, College of Veterinary Medicine, University of Tennessee-Knoxville, Knoxville, TN, United States
[2] Department of Large Animal Clinical Sciences, College of Veterinary Medicine, University of Tennessee-Knoxville, Knoxville, TN, United States
[3] Research Computing Support, University of Tennessee Office of Innovative Technologies, University of Tennessee-Knoxville, Knoxville, TN, United States

## ABSTRACT

**Background:** The majority of published reference intervals for hematology and clinical biochemistry in pigs are generated from a sample group that is demographically different from companion pigs, and as such may not be transferable. The goals of this study were to provide reference intervals generated from sexually mature companion pigs and to compare results based on age group, breed, and reproductive status. Reference intervals are ideally generated in the same laboratory in which patient samples are measured, since there is often bias in values generated from different instruments, but the cost and time commitment required to produce reference intervals may be prohibitive. If so, published reference intervals may be used cautiously as guidelines for interpretation.

**Methods:** Complete blood count (CBC) and plasma biochemistry data were generated using the ADVIA 2120 hematology analyzer and Cobas c501 chemistry analyzer on blood samples collected from 94 sexually mature, clinically healthy companion pigs housed mostly in eastern Tennessee over a 5-year period. The majority (90/94) of samples were collected after sedation or general anesthesia. The age range of the reference sample group was 5 months to 11 years, including <1-year-old ($n = 26$), 1–2 years old ($n = 26$), and >2-years-old ($n = 42$). Reproductive status included intact females ($n = 46$), spayed females ($n = 9$), intact males ($n = 15$), and castrated males ($n = 24$). Breeds were predominantly Vietnamese potbellied mini pigs, American mini pigs, and mixed breed pigs.

**Results:** Reference intervals are provided for routine CBC and plasma biochemistry values. The <1-year-old pigs were excluded from reference interval calculation for some values because their results were significantly different from pigs >1-year-old. These included red blood cell concentration, mean cell volume, mean cell hemoglobin, platelet count, mean platelet volume, lymphocyte concentrations by both automated and manual methods, and total protein by refractometry. Few significant differences were observed based on breed or reproductive status.

**Discussion:** Age, breed, and reproductive status can affect some hematology and biochemistry results in companion pigs. If companion pig reference intervals are not available from the laboratory in which patient samples are measured, these published

Corresponding author
Deanna M. W. Schaefer,
dschaefe@utk.edu

reference intervals may provide guidance for interpretation, although some methodologic variances are likely.

# INTRODUCTION

Companion pigs are becoming increasingly popular (*Curnutte, 2014*), and as such, routine blood testing including complete blood count (CBC) and clinical biochemistry testing is performed more frequently. In the clinical pathology laboratory of the University of Tennessee Veterinary Medical Center (UTVMC), for example, the number of plasma biochemistry profiles performed on porcine samples increased over threefold in the decade spanning 2010–2019 compared to 2000–2009. Because the results of these tests depend on factors such as species, breed, age, sex, and husbandry conditions, the recommendation of the American Society for Veterinary Clinical Pathology (ASVCP) is to generate population-based reference intervals from a reference sample group with similar demographics to the animal population in which they will be used (*Friedrichs et al., 2012*). These ASVCP guidelines also provide specific recommendations regarding methods for determining and reporting reference intervals, including statistical methods, based on the number of individuals in the reference group. However, the majority of published porcine reference intervals are either not compliant with these recommendations or are from sample groups dissimilar to clinical porcine patients, such as commercial production pigs, research pigs, sexually immature pigs, or pregnant or lactating sows (*Brockus et al., 2005*; *Dimitrakakis et al., 2022*; *Li et al., 2021*; *Perri et al., 2017*; *Radin, Weiser & Fettman, 1986*; *Verheyen et al., 2007*; *Yeom et al., 2012*).

The main objective of this study was to provide CBC and clinical biochemistry reference intervals for sexually mature companion pigs using ASVCP guidelines. A secondary objective was to compare results based on age group, breed, and reproductive status.

# MATERIALS AND METHODS

## Reference sample group

Blood was collected from healthy companion pigs that had reached sexual maturity and were presented to the UTVMC large animal hospital for routine procedures such as hoof trimming, tusk trimming, vaccinations, and/or microchip placement; or for elective surgeries such as ovariohysterectomy, ovariectomy, or castration. They were deemed healthy based on history and physical exam findings and were fasted for 8–10 h prior to blood collection.

## Blood collection

Blood samples were collected from 2019 to 2024 according to approved University of Tennessee IACUC protocols (UT IACUC protocol numbers 22980914 for 2019–2022 and 29470123 for 2023–2024). Whole blood was collected into a tube containing

ethylenediamine tetraacetic acid anticoagulant (EDTA) for the CBC and into a lithium heparin tube for clinical biochemistry. The majority of samples (90/94) were collected while the pig was sedated or anesthetized for clinical procedures, with the most common drugs being xylazine, midazolam, ketamine, and isoflurane, according to clinician preference. The venipuncture site was also selected according to clinician preference, with the most common sites being the cranial vena cava or lateral saphenous vein. Blood tubes were submitted to the UTVMC clinical pathology laboratory within 30 min of collection.

## Analytical methods

The CBC and clinical biochemistry panels were performed using an ADVIA 2120 (Siemens Healthcare Diagnostics, Tarrytown, NY, USA) and a Cobas c501 (Roche Diagnostics, Rotkreuz, Switzerland), respectively, operated by a trained medical laboratory scientist or laboratory assistant according to the UTVMC clinical pathology laboratory standard operating procedures. A summary of analyzer methods is provided in Table S1. To ensure optimal performance of laboratory instrumentation, daily quality control procedures and periodic scheduled preventive maintenance are performed for each analyzer, and the UTVMC clinical pathology laboratory participates in external quality assurance programs for each instrument. No analytical performance issues were identified during measurement of the study samples.

The majority of values reported for the CBC and chemistry panels were generated by the automated methods of the analyzers. Additionally, a white blood cell (WBC) percentage differential was determined by the manual method, that is through microscopic enumeration of at least one hundred consecutive leukocytes by a trained medical laboratory scientist. This percentage differential was used along with the automated total WBC concentration to calculate absolute concentrations of each leukocyte. Results for both the automated differential and the manual differential were recorded. Each blood smear was also microscopically reviewed for platelet clumping. The automated platelet concentration from samples with platelet clumps was excluded from analysis only if the platelet concentration was identified statistically as an outlier low value. The EDTA whole blood sample was additionally used to estimate total protein by refractometry, and refractometry was also used to estimate fibrinogen by heat precipitation according to the laboratory standard operating procedure and published descriptions (*George, 2001*).

## Statistical methods

Statistical software used for reference interval determination included MedCalc® version 22.023 (MedCalc Software Ltd., Ostend, Belgium) and Reference Value Advisor Freeware v2.1 add-in for Excel (*Geffre et al., 2011*). Outliers were identified by the Tukey method and by visual inspection of histograms, then were inspected for transcriptional errors or to justify exclusion, such as samples having marked lipemia. Unless there was a specific justification to remove the data, outliers were included in the reference interval calculation. Distribution as Gaussian or non-Gaussian was determined by "the Anderson-Darling method" and by visual inspection of histogram results. If this initial evaluation was not symmetrical, Box-Cox transformation was applied. If data remained non-symmetric after

transformation, the data were considered non-Gaussian. Reference intervals were calculated using the methods recommended by ASVCP for sample sizes between 40 and 120. Namely, if data had Gaussian distribution, before or after Box-Cox transformation, reference limits were based on the robust method. If non-Gaussian, the non-parametric method was used. The 90% confidence intervals of the reference limits were determined using a bootstrap method. As an estimate of the uncertainty of the reference limits, each resulting reference interval was examined to determine whether the width of confidence intervals for the lower and upper limits met the Boyd and Harris guidelines that they did not exceed 0.2 times the width of the reference interval (*Friedrichs et al., 2012*).

To evaluate the effects of age on reference intervals, pigs were partitioned into three groups, those 5–11 months of age, those 1–2 years old, and those greater than 2 years old. This statistical evaluation was performed using SAS version 9.4, release TS1M8 (SAS Institute Inc., Cary, NC, USA). The normality of the data was evaluated using the Shapiro-Wilk W and QQ plot, with normally distributed data evaluated using a one-way ANOVA and non-normally distributed variables analyzed by a one-way ANOVA on rank transformation. Holm–Bonferroni method was used to adjust the p-values for multiple dependent variables. If significant differences were detected based on age, the reference interval was recalculated after excluding pigs <1 year of age. Similar statistical methods were used to detect significant differences based on sex after partitioning into four groups: intact female, spayed female, intact male, and castrated male, and based on breed after partitioning into Vietnamese potbellied pig, American mini pig, or mixed breed pig. The least squares means were computed, and *post hoc* paired comparison was adjusted by Tukey Karmer method. All statistical assumptions of normality and equality of variances were met. $P < 0.05$ was considered significant.

## RESULTS

### Reference sample group

Blood was collected from 94 pigs during the study period, with each pig being sampled only one time. Blood collections occurred during spring ($n = 32$), summer ($n = 27$), fall ($n = 16$), and winter ($n = 19$). The majority of pigs were housed in east Tennessee ($n = 90$), with low numbers from western North Carolina ($n = 3$) or northern Virginia ($n = 1$). Age ranged from 5 months to 11 years (median 2 years). Intact males were significantly younger than other sex groups ($P = 0.025$, median 1 year, range 7 months to 3 years). Vietnamese potbellied pigs were significantly older than other breeds ($P = 0.0002$, median 3 years, range 9 months to 11 years). Additional demographic data for the group is provided in Table 1.

### Reference intervals

Plasma biochemistry panels were available for all 94 pigs. CBC results were available for 92 pigs. Blood in the EDTA tubes from two pigs was clotted, precluding CBC analysis. Reference intervals for the RBC and platelet portions of the CBC are provided in Table 2, while WBC portions are provided in Table 3, including both the automated WBC differential provided by the analyzer and the manual differential generated by microscopic

**Table 1 Reference sample group demographic data.**

| | Description | Number of pigs in group |
|---|---|---|
| Age | <1-year-old | 26 |
| | 1–2 years-old | 26 |
| | >2 years-old | 42 |
| Sex | Intact female | 46 |
| | Spayed female | 9 |
| | Intact male | 15 |
| | Castrated male | 24 |
| Breed | Vietnamese potbellied mini pig | 41 |
| | Mixed breed | 39 |
| | American mini pig | 13 |
| | Meishan pig | 1 |

blood smear review. Refractometric protein measurements are in Tables 4–6 display clinical biochemistry reference intervals. Results are reported in accordance with the recommendations of the ASVCP Quality Assurance and Laboratory Standards Committee, including the reference interval, sample size, mean, standard deviation, median, minimum, maximum, 90% confidence intervals of the reference limits, and the statistical method used (*ASVCP, 2020*; *Friedrichs et al., 2012*).

## Excluded values

Due to the prolonged study period and due to logistical issues, not all pigs had results available for every value. For example, plasma iron measurement was not added to the routine large animal chemistry profile at UTVMC until 2020, there was insufficient funding to measure triglycerides, cholesterol, or reticulocytes in all samples, and occasional values were inadvertently omitted from the analyses of some samples. Additionally, pigs less than 1 year of age were omitted from the calculation of reference intervals for values in which their results were significantly different from other age groups.

Guidelines from ASVCP emphasize that values at the extremities of distribution should not be eliminated from reference interval calculations without justification (*Friedrichs et al., 2012*). As such, values identified as statistical outliers were retained in most cases. However, outliers that were excluded from reference interval calculations included a low platelet count ($14 \times 10^9$/L) and low plateletcrit (0.01%) from one sample with numerous large platelet clumps identified on the blood smear. While the majority of blood smears had at least rare small platelet clumps identified, this was the only sample statistically identified as an outlier for platelet concentration, so was the only excluded value. In two pigs, high results for AST (189 and 285 U/L) and CK (6,999 U/L and an unreadable measurement) were excluded, presumably indicating muscle injury during transport or sample collection. Lastly, two high GGT measurements were excluded (443 and 874 U/L), which were from the two most lipemic samples in the study. The lipemia indices for these samples were 166 and 308, whereas this index in the remaining samples ranged from 5 to

**Table 2 Hematology reference intervals for companion pigs, red blood cells and platelets.**

| Value (Units) | REFERENCE INTERVAL | N | Mean | SD | Median | Min | Max | LRL 90% CI | URL 90% CI | Stat[a] |
|---|---|---|---|---|---|---|---|---|---|---|
| RBC ($10^{12}$/L) | 2.70–7.67 | 66 | 5.01 | 2.44 | 5.01 | 2.25 | 8.16 | [2.25–3.45] | [7.01–8.16] | NG |
| HGB (g/dl) | 7.8–16.0 | 92 | 10.6 | 1.8 | 10.6 | 6.1 | 17.6 | [6.1–8.3] | [13.2–17.6] | NG |
| HCT (%) | 22.2–43.7 | 92 | 30.8 | 5.5 | 30.1 | 17.3 | 50.9 | [21.1–23.5] | [40.1–47.4] | T |
| MCV (fL) | 52.0–78.9 | 66 | 62.2 | 28.6 | 62.0 | 51.7 | 82.7 | [51.7–53.8] | [70.9–82.7] | NG |
| MCH (pg) | 18.4–26.9 | 66 | 21.6 | 9.9 | 21.6 | 18.2 | 27.2 | [18.2–18.7] | [24.0–27.2] | NG |
| MCHC (g/dl) | 32.1–35.9 | 92 | 34.5 | 1.01 | 34.6 | 30.9 | 36.9 | [30.9–32.8] | [35.9-36.9] | NG |
| RDW (%) | 13.5–23.4 | 92 | 16.2 | 2.3 | 16.0 | 13.4 | 30.8 | [13.4–13.8] | [18.6–30.8] | NG |
| Retic ($10^9$/L) | NA | 20 | 50.2 | 27.7 | 50.2 | 18.7 | 131.4 | NA | NA | NA |
| nRBCs ($10^9$/L) | 0–0.2 | 92 | 0 | 0.04 | 0 | 0 | 0.2 | [0–0] | [0.1–0.2] | NG |
| Platelets ($10^9$/L) | 145–504 | 65 | 326 | 167 | 339 | 147 | 545 | [119–178] | [471–534] | G |
| Plateletcrit (%) | 0.20–0.47 | 91 | 0.33 | 0.08 | 0.32 | 0.19 | 0.67 | [0.19–0.23] | [0.45–0.67] | NG |
| MPV (fL) | 6.9–16.7 | 66 | 10.3 | 5.0 | 9.9 | 6.5 | 16.9 | [6.5–7.5] | [14.2–16.9] | NG |

Notes:
Abbreviations: HCT, hematocrit; HGB, hemoglobin concentration; LRL 90%CI, 90% confidence interval of the lower reference limit; Max, maximum value; MCH, mean cell hemoglobin; MCHC, mean cell hemoglobin concentration; MCV, mean cell volume; Min, minimum value; MPV, mean platelet volume; N, sample size; NA, not applicable because sample size is too small to calculate a reference interval; nRBCs, nucleated red blood cell concentration; RBC, red blood cell concentration; RDW, red cell distribution width; Retic, reticulocyte concentration; SD, standard deviation; URL 90%CI, 90% confidence interval of the upper reference limit.
[a] Stat indicates the distribution of data and statistical method.
G, Gaussian distribution, robust method; T, Gaussian after transformation, robust method; NG, non-Gaussian, non-parametric method.

74. The lipemia index is a numeric value provided by the chemistry analyzer used in this study. It is based on absorption of light at various wavelengths and correlates to turbidity caused by lipemia. A lipemia index of greater than 100–120 has been correlated to the visual equivalent of moderate to marked lipemia (*eClinPath, 2024*; *Getahun et al., 2019*; *Lim & Cha, 2017*).

## Partitioning by age

For the CBC, the following values were significantly lower in <1 year old pigs compared to at least one of the remaining age groups: mean cell volume (MCV, $P < 0.001$), mean cell hemoglobin (MCH, $P < 0.0001$), and mean platelet volume (MPV, $P < 0.001$). Additionally, the following were significantly higher in <1 year old pigs: red blood cell concentration (RBC, $P = 0.031$), platelet concentration ($P < 0.001$), and lymphocyte concentration by both automated ($P < 0.001$) and manual ($P < 0.001$) methods. For a few of these hematology values, the 1–2 year old pigs were also significantly different from the

**Table 3 Hematology reference intervals for companion pigs, white blood cells.**

| Value ($10^9$/L) | REFERENCE INTERVAL | N | Mean | SD | Median | Min | Max | LRL 90% CI | URL 90% CI | Stat[a] |
|---|---|---|---|---|---|---|---|---|---|---|
| WBC | 4.3–19.5 | 92 | 9.5 | 3.9 | 8.9 | 3.3 | 28.7 | [3.3–4.8] | [16.9–22.3] | T |
| Neutr auto | 1.1–13.3 | 92 | 4.4 | 3.2 | 3.7 | 0.7 | 21.7 | [0.9–1.3] | [10.6–16.5] | T |
| Lymph auto | 1.9–6.3 | 66 | 3.9 | 1.1 | 3.9 | 1.6 | 7.6 | [1.6–2.2] | [5.9–6.9] | T |
| Mono auto | 0.1–0.9 | 92 | 0.4 | 0.2 | 0.4 | 0.1 | 1.1 | [0.1–0.2] | [0.8–1.1] | NG |
| Eos auto | 0–1.0 | 92 | 0.2 | 0.3 | 0.2 | 0 | 2.1 | [0–0] | [0.7–1.5] | T |
| Baso auto | 0–0.1 | 92 | 0 | 0 | 0 | 0 | 0.2 | [0–0] | [0.1–0.2 | NG |
| LUC auto | 0–0.4 | 92 | 0.1 | 0.1 | 0.1 | 0 | 0.4 | [0–0] | [0.4–0.4] | NG |
| Neutr manual | 1.0–14.6 | 92 | 4.6 | 3.5 | 3.9 | 0.7 | 23.5 | [0.8–1.2] | [11.7–17.8] | T |
| Band manual | 0–0.2 | 92 | 0 | 0 | 0 | 0 | 0.2 | [0–0] | 0.1–0.2] | NG |
| Lymph manual | 1.5–6.6 | 66 | 3.6 | 1.3 | 3.5 | 1.3 | 7.2 | 1.3–1.7] | [5.8–7.1] | NG |
| Mono manual | 0–1.4 | 92 | 0.5 | 0.3 | 0.4 | 0 | 1.7 | [0–0.1] | [1.7–1.7] | NG |
| Eos manual | 0–1.7 | 92 | 0.2 | 0.4 | 0.2 | 0 | 2.5 | [0–0] | [0.7–2.5] | NG |
| Baso manual | 0–0.5 | 92 | 0.1 | 0.1 | 0.1 | 0 | 0.7 | [0–0] | [0.2–0.7] | NG |

Notes:

Abbreviations: Band manual, band neutrophil concentration by microscope; Baso auto, basophil count by analyzer; Baso manual, basophil concentration by microscope; Eos auto, eosinophil concentration by analyzer; Eos manual, eosinophil concentration by microscope; LRL 90%CI, 90% confidence interval of the lower reference limit; LUC auto, large unclassified cells by analyzer (usually includes reactive lymphocytes and some monocytes); Lymph auto, lymphocyte concentration by analyzer; Lymph manual, lymphocyte concentration by microscope; Max, maximum value Min, minimum value; Mono auto, monocyte concentration by analyer; Mono manual, monocyte concentration by microscope; N, sample size; Neutr auto, neutrophil concentration by analyzer; Neutr manual, segmented neutrophil concentration by microscope; SD, standard deviation; URL 90%CI, 90% confidence interval of the upper reference limit; WBC, total white blood cell concentration.
[a] Stat indicates the distribution of data and statistical method.
G, Gaussian distribution, robust method; T, Gaussian after transformation, robust method; NG, non-Gaussian, non-parametric method.

**Table 4 Refractometric protein reference intervals for companion pigs.**

| Value (Units) | REFERENCE INTERVAL | N | Mean | SD | Median | Min | Max | LRL 90%CI | URL 90%CI | Stat[a] |
|---|---|---|---|---|---|---|---|---|---|---|
| Total protein (g/dl) | 5.7–8.7 | 66 | 7.3 | 3.3 | 7.3 | 5.6 | 9.0 | [5.6–6.1] | [8.4–9.0] | NG |
| Fibrinogen heat prec (mg/dl) | 0–600 | 68 | 182 | 157 | 100 | 0 | 700 | [0–0] | [427–700] | NG |

Notes:

Abbreviations: Fibrinogen heat prec, estimated fibrinogen concentration by heat precipitation method;N, sample size; LRL 90%CI, 90% confidence interval of the lower reference limit; Max, maximum value; Min, minimum value; SD, standard deviation; URL 90%CI, 90% confidence interval of the upper reference limit.
[a] Stat indicates the distribution of data and statistical method.
G, Gaussian distribution, robust method; T, Gaussian after transformation, robust method; NG, non-Gaussian, non-parametric method.

pigs greater than 2 years old, specifically MCV, MCH, platelet concentration and MPV. The youngest age group also had significantly lower results for total protein by refractometry ($P < 0.001$). Total protein, albumin, and globulins on the chemistry panel also had a trend toward being lower in the youngest group, but differences did not reach statistical significance.

For all of these values, the reference intervals provided in Table 1 include only pigs that are at least 1-year-old. Calculation of reference intervals for subgroups after partitioning is only recommended if each of the subgroups contains at least 40 individuals (*Friedrichs et al., 2012*). For this reason, partitioned reference intervals were not calculated separately for each of the three age groups. Instead, graphical representation of results from

**Table 5  Plasma biochemistry reference intervals for companion pigs, part 1.**

| Value (Units) | REFERENCE INTERVAL | N | Mean | SD | Median | Min | Max | LRL 90% CI | URL 90%CI | Stat[a] |
|---|---|---|---|---|---|---|---|---|---|---|
| Tot Prot (g/dl) | 5.4–8.2 | 94 | 6.8 | 0.7 | 6.8 | 4.7 | 8.4 | [5.2–5.6] | [8.0–8.4] | G |
| Albumin (g/dl) | 3.1–5.3 | 94 | 4.2 | 0.5 | 4.2 | 2.8 | 5.4 | [2.8–3.3] | [5.0–5.4] | NG |
| Globulins (g/dl) | 1.0–4.2 | 94 | 2.6 | 0.7 | 2.7 | 0.7 | 3.9 | [0.7–1.3] | [3.7–3.9] | NG |
| A:G ratio (g/dl) | 0.9–4.2 | 94 | 1.8 | 1.0 | 1.5 | 0.8 | 7.8 | [0.8–1.0] | [3.3–7.8] | NG |
| Glucose (mg/dl) | 35–197 | 94 | 101 | 34 | 96 | 29 | 227 | [29–60] | [166–227] | NG |
| Chol (mg/dl) | 28–153 | 61 | 77 | 32 | 68 | 23 | 182 | [24–33] | [134–171] | T |
| Trig (mg/dl) | 8.4–114.5 | 65 | 39.5 | 26.1 | 31.6 | 6.7 | 129.8 | [6.6–10.6] | [90.1–140.6] | T |
| AST (U/L) | 13–94 | 92 | 33 | 19 | 30 | 11 | 130 | [11–15] | [71–130] | NG |
| SDH (U/L) | 0–8.0 | 93 | 1.9 | 2.5 | 1.1 | 0 | 17.2 | [0–0] | [6.5–17.2] | NG |
| GGT (U/L) | 34–141 | 92 | 65 | 29 | 59 | 27 | 228 | [27–38] | [107–228] | NG |
| Bilirubin (mg/dl) | 0–0.4 | 94 | 0.1 | 0.1 | 0.1 | 0 | 0.4 | [0–0] | [0.3–0.4] | NG |
| CK (U/L) | 209–3,850 | 92 | 905 | 815 | 716 | 173 | 4,953 | [173–249] | [2,411–4,953] | NG |

Notes:
Abbreviations: A:G ratio, albumin to globulin ratio; AST, aspartate transaminase; Chol, cholesterol; CK, creatine kinase; GGT, gammaglutamyl transferase; LRL 90%CI, 90% confidence interval of the lower reference limit; Max, maximum value; Min, minimum value; N, sample size; SD, standard deviation; SDH, sorbitol dehydrogenase; Tot Prot, total protein; Trig, triglycerides; URL 90%CI, 90% confidence interval of the upper reference limit.
[a] Stat indicates the distribution of data and statistical method.
G, Gaussian distribution, robust method; T, Gaussian after transformation, robust method; NG, non-Gaussian, non-parametric method.

subgroups is provided in Figs. 1 and 2 for values in which a statistically significant difference was noted between subgroups.

## Partitioning by sex and breed

When grouped by sex as intact female, intact male, spayed female, or castrated male, the only significant differences were that intact males had a higher neutrophil count by automated ($P = 0.010$) and manual ($P = 0.016$) methods (Fig. 3). A few significant differences were also noted between breeds (Fig. 4). The single Meishan pig was excluded from these comparisons. Vietnamese potbellied pigs had a lower manual lymphocyte count ($P = 0.001$) than the other two breeds. Total protein on the chemistry panel and total protein estimated by refractometry were both significantly different among the breeds ($P = 0.013$ and $P = 0.011$, respectively), with American mini pigs having the lowest values.

**Table 6 Plasma biochemistry reference intervals for companion pigs, part 2.**

| Value (Units) | REFERENCE INTERVAL | N | Mean | SD | Median | Min | Max | LRL 90% CI | URL 90%CI | Stat[a] |
|---|---|---|---|---|---|---|---|---|---|---|
| Urea (mg/dl) | 2–18 | 94 | 10 | 4 | 10 | 2 | 20 | [2–5] | [16–20] | NG |
| Creatinine (mg/dl) | 0.6–2.3 | 94 | 1.2 | 0.4 | 1.1 | 0.4 | 2.5 | [0.4–0.7] | [1.9–2.5] | NG |
| Phos (mg/dl) | 4.0–8.0 | 94 | 5.6 | 1.2 | 5.2 | 3.5 | 12.3 | [3.5–4.2] | 7.3–12.3] | NG |
| Calcium (mg/dl) | 9.1–10.9 | 94 | 10.0 | 0.5 | 10.0 | 8.9 | 11.1 | [8.9–8.2] | [10.8–11.1] | G |
| Magn (mmol/l) | 0.7–1.0 | 93 | 0.8 | 0.1 | 0.8 | 0.7 | 1.1 | [0.7–0.7] | [1.0–1.1] | NG |
| Iron (µg/dl) | 51.5–208.7 | 42 | 113.4 | 39.2 | 106.7 | 29.9 | 272.4 | [41.8–66.3] | [168.3–248.9] | T |
| Sodium (mmol/l) | 135–146 | 93 | 141 | 3 | 141 | 135 | 150 | [134–136] | [145–147] | G |
| Potassium (mmol/l) | 3.4–4.9 | 93 | 4.0 | 0.4 | 3.9 | 3.3 | 5.8 | [3.4–3.5] | [4.7–5.2] | T |
| Chloride (mmol/l) | 96–107 | 93 | 101 | 3 | 101 | 90 | 107 | [95–97] | [106–108] | G |
| TCO2 (mmol/l) | 19.1–34.2 | 93 | 28.1 | 4.3 | 28.9 | 8.1 | 36.4 | [16.0–21.5] | [33.4–34.9] | T |
| Anion Gap (mmol/l) | 10–36 | 93 | 15 | 6 | 14 | 10 | 48 | [10–10] | [30–48] | NG |

Notes:
Abbreviations: LRL 90%CI, 90% confidence interval of the lower reference limit; Magn, magnesium; Max, maximum value; Min, minimum value; N, sample size; Phos, phosphorus; SD, standard deviation; TCO2, total carbon dioxide/bicarbonate; URL 90%CI, 90% confidence interval of the upper reference limit.
[a] Stat indicates the distribution of data and statistical method.
G, Gaussian distribution, robust method; T, Gaussian after transformation, robust method; NG, non-Gaussian, non-parametric method.

## Confidence intervals

There is inherent uncertainty regarding the accuracy of all reference limits. For that reason, ASVCP recommends reporting the 90% confidence intervals of each limit. Furthermore, wide confidence intervals indicate reduced certainty in the reference limits, with guidelines that the ratio of the width of the confidence interval for the lower or upper limit to the width of the reference interval ideally should not exceed 0.2 (*Friedrichs et al., 2012*). A ratio higher than 0.2 indicates that patient values near the reference limit may or may not indicate pathologic change. This ratio exceeded 0.2 for the upper reference limit of the majority of CBC and chemistry values in this study and for the lower reference limit in a few values (Table S2).

## DISCUSSION

The objectives of this study were to provide hematologic and biochemical reference intervals for companion pigs that are compliant with ASVCP guidelines and to evaluate for differences based on age, sex, and breed. Reference intervals derived from healthy individuals of a species are used to identify potentially pathologic abnormalities on CBC

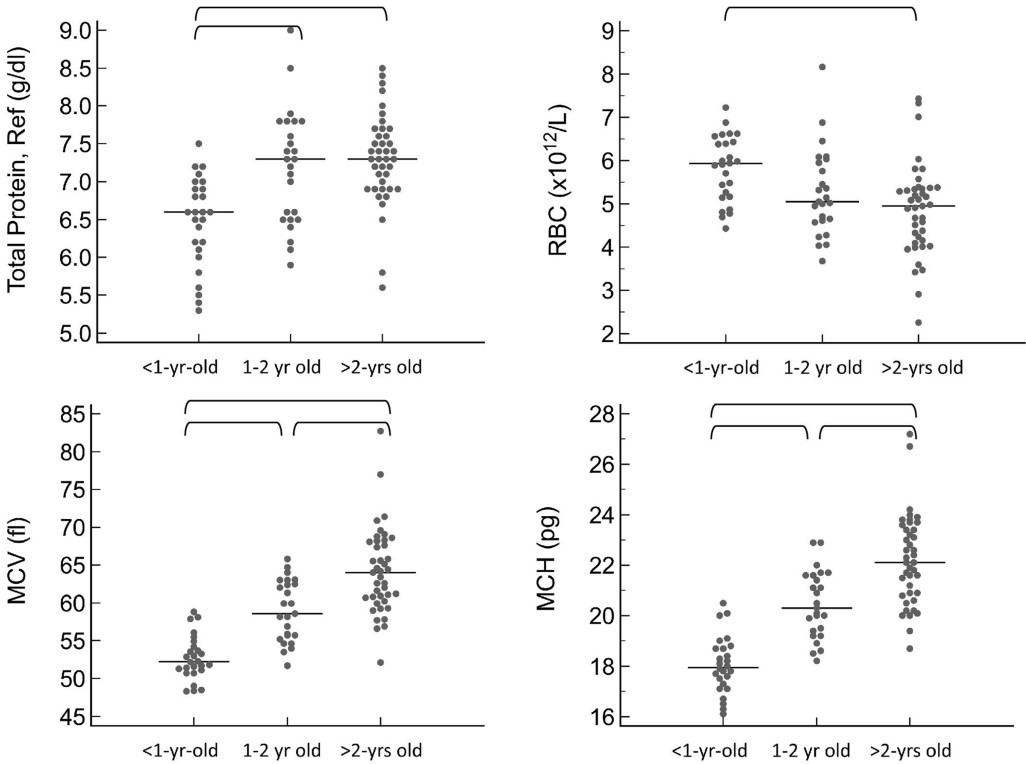

**Figure 1 Results for significantly different protein and red blood cell values on the CBC, grouped by age as <1-year old, 1–2 years old, and >2-years old.** Horizontal straight lines indicate the median for each group. Brackets denote significant differences between groups. Abbreviations: MCH, mean cell hemoglobin; MCV, mean cell volume; RBC, red blood cell concentration; Total Protein Ref, total protein estimated by refractometry.

and biochemistry panels from clinical patients. These reference intervals are ideally determined *de novo* in each laboratory to account for bias between instruments and methods. This can be time-consuming and expensive to perform for all species seen in veterinary hospitals, though. As such, for some species, clinicians must rely on published reference intervals to interpret laboratory data. However, many published reference intervals for pigs are derived from sample groups that do not mimic our clinical patient population of companion pigs, but rather are derived from research colonies, commercial swine, or juvenile pigs (*Dimitrakakis et al., 2022*; *Li et al., 2021*; *Perri et al., 2017*; *Radin, Weiser & Fettman, 1986*; *Verheyen et al., 2007*).

Comparison of the methods and results of our study to those of six other porcine reference interval publications is provided in Tables S3–S5. For most values, our reference intervals are similar to previously published values, but some differences are noted. However, the sample groups for these studies varied in breed, age, and number of individuals, and there were also differences in analytical and statistical methods, all of which may contribute to differences in results. For example, the lower reference limit of our lymphocyte reference interval is lower than those published for juvenile pigs (*Dimitrakakis et al., 2022*; *Li et al., 2021*), but it is similar to that published for adult potbellied pigs (*Brockus et al., 2005*), suggesting an effect of animal age. However,

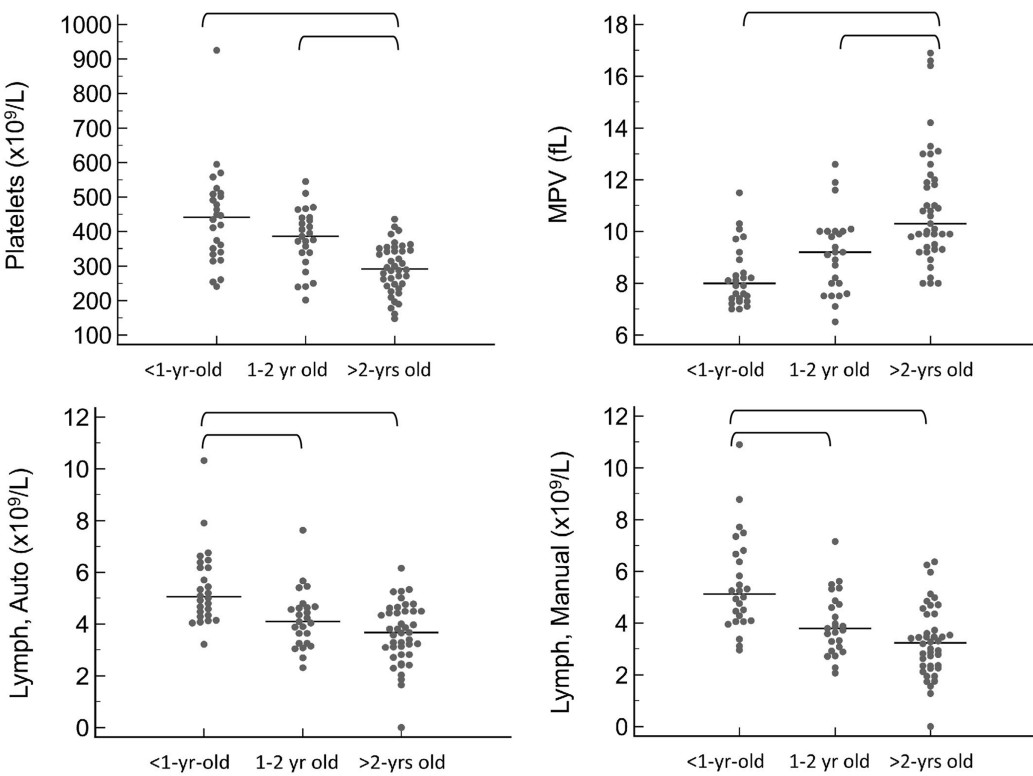

**Figure 2 Results for significantly different platelet and white blood cell values on the CBC, grouped by age as <1-year old, 1–2 years old, and >2-years old.** Horizontal straight lines indicate the median for each group. Brackets denote significant differences between groups. Abbreviations: Lymph Auto, lymphocyte concentration by analyzer; Lymph Manual, lymphocyte concentration by microscope; MPV, mean platelet volume.

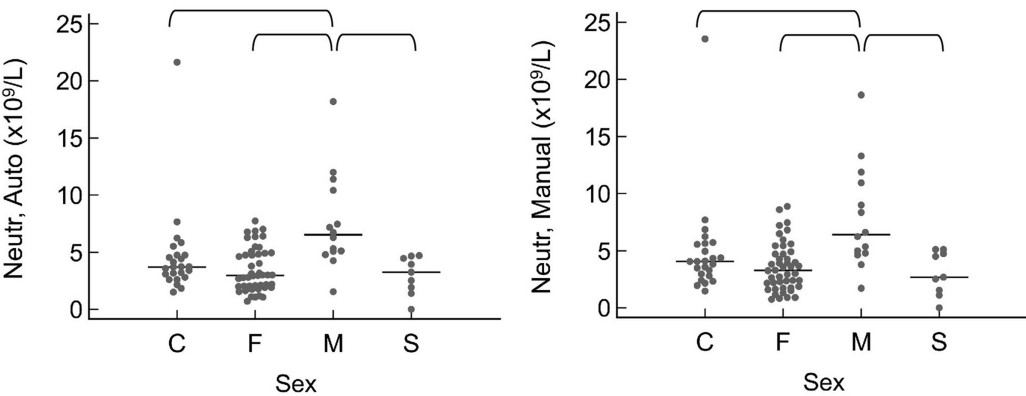

**Figure 3 Results for significantly different values, grouped by sex as C (castrated male), F (intact female), M (intact male), and S (spayed female).** Horizontal straight lines indicate the median for each group. Brackets denote significant differences between groups. Abbreviations: Neutr Auto, neutrophil concentration by analyzer; Neutr Manual, neutrophil concentration by microscope.

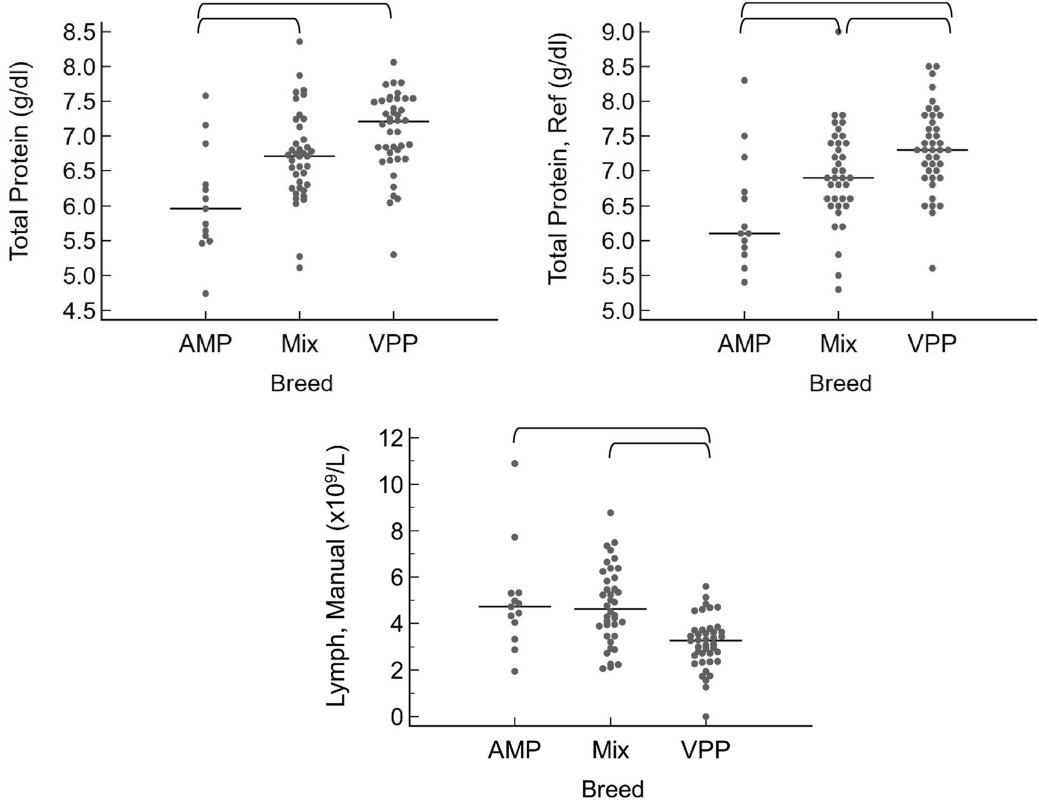

**Figure 4 Results for significantly different values, grouped by breed as AMP (American mini pig), Mix (mixed breed pig), and VPP (Vietnamese potbellied pig).** Horizontal straight lines indicate the median for each group. Brackets denote significant differences between groups. Abbreviations: Lymph Manual, lymphocyte concentration by microscope; Total Protein Ref, total protein estimated by refractometry.

compared to the same study of adult potbellied pigs, our chloride reference interval was lower, which might indicate differences in analytical methodology.

Benefits of the current study include that the reference intervals were derived from client-owned healthy companion pigs seen at the UTVMC, so reflect companion pig patients, and were reported using ASVCP recommendations for a sample group of 40–120 individuals. Additionally, the reference sample group included a mix of ages, sexes, and breeds, allowing comparison of results based on these factors. Finally, some of the values reported here are not included in most prior studies, such as WBC differentials by both automated and manual methods, SDH, magnesium, and iron.

The majority of differences noted between pigs less than or greater than 1 year of age have also been identified in other species. The 5 to 11-month-old pigs had lower MCV and MCH compared to pigs at least 1 year of age, which was also reported in 6 to 12-month-old specific pathogen free minipigs compared to those at least 1 year of age (*Yeom et al., 2012*). Red cells have been reported to be smaller in younger animals of many species compared to adults, with those smaller red cells contributing to decreases in both MCV and MCH (*Harper et al., 2003*; *Monke et al., 1998*; *Muñoz et al., 2012*; *Stockham & Scott, 2008a*). One proposed mechanism for production of smaller red cells in young rapidly-growing animals

is that they may have low iron stores due to an all-milk diet during the neonatal period (*Brockus, 2011*). Iron deficiency is a common cause of production of microcytic erythrocytes (*Li et al., 2021*; *Stockham & Scott, 2008a*). It is possible that the youngest age group in our study either still had low iron stores contributing to decreases in red cell size, or had gained adequate iron stores by the time of sample collection but had not yet replaced all neonatal red cells with normocytic cells. Notably, the serum iron was not significantly lower in the youngest pigs of our study nor of the study of specific pathogen free minipigs (*Yeom et al., 2012*), supporting adequate iron stores by the time of sample collection. However, iron was only measured in 42 out of the 94 pigs in our study, including 11 pigs <1 year of age, 12 pigs between 1 and 2 years, and 19 pigs >2-years-old, which may have impacted the ability to detect age-related differences. Interestingly, the RBC concentration in 5 to 11-month-old pigs was higher than that of older pigs, suggesting that decreased iron stores, if present, were not measurably impairing erythropoiesis. The cause for the higher RBC count in these young pigs cannot be confirmed but may indicate either increased erythropoiesis or a relative increase due to excitement-mediated splenic contraction. Higher RBC concentration is not a commonly reported feature of young animals.This trend was noted in at least one publication in horses, but not in specific pathogen free minipigs (*Muñoz et al., 2012*; *Yeom et al., 2012*). Pigs less than 1 year of age also had higher lymphocyte concentrations than the older pig groups, which has been reported to occur commonly in other species, possibly either due to increased immune stimulation or physiologic leukocytosis (*Barton & Hart, 2020*; *Harper et al., 2003*; *Monke et al., 1998*; *Panousis et al., 2018*; *Stockham & Scott, 2008b*).

Additionally, younger pigs had increased platelet concentration and decreased MPV compared to older pigs. However, plateletcrit was not significantly different based on age, presumptively due to the summative effects of higher platelet numbers but smaller platelet size. Higher platelet numbers have been reported in younger calves and humans, but in these reports the MPV is either similar to or higher than that of adults (*Jeon, 2022*; *Novak, Tschantz & Krill, 1987*; *Panousis et al., 2018*). The mechanism for these higher platelet counts has not been proven, but in pediatric humans is suspected to be promoted by accelerated thrombopoiesis due to increased circulating thrombopoietin concentrations relative to adults (*Jeon, 2022*). It is possible that this phenomenon may also occur in pigs, but accelerated thrombopoiesis would be expected to promote an increase in MPV rather than a decrease (*Stockham & Scott, 2008c*). Therefore, a mechanism for the age-related changes in platelet concentration and MPV in these pigs is uncertain. Lastly, the lower protein concentration noted in the younger pigs has been previously reported in other species (*Barton & Hart, 2020*; *Harper et al., 2003*; *Monke et al., 1998*; *Muñoz et al., 2012*). In very young animals, this can occur due to variable absorption of colostral immunoglobulins, but this would be an unlikely explanation for pigs of the ages included in our study. A more plausible theory for our study population, given that there was a trend toward decreases in both albumin and globulin in the youngest group, is that non-immunoglobulin protein concentrations may increase as animals mature due to increasing hepatic protein synthetic capability (*Muñoz et al., 2012*).

Notably, there were a few values in which the differences were also significant between 1 to 2-year-old pigs and pigs greater than 2-years-old, specifically MCV, MCH, platelet concentrations, and MPV, suggesting that some values may not reach adult levels until pigs are older than 1 year. These findings highlight the importance of accounting for patient age when interpreting laboratory results.

In contrast to age groups, there were few significant differences when pigs were partitioned based on sex or breed. Intact males had significantly increased neutrophil concentrations, with possible explanations being increased likelihood of glucocorticoid or epinephrine responses during handling or of subclinical inflammatory disease. There were no other laboratory changes supportive of these conditions in the intact male pigs, such as changes in lymphocyte or band neutrophil concentrations, but that does not exclude these possibilities. Intact males were significantly younger than the other sex groups, but neutrophils concentrations did not differ based on age, so a cause for the neutrophil differences is unclear. For breed groups, a mild but significant difference was noted in the lymphocyte concentration by the manual method, with Vietnamese potbellied pigs having slightly lower values. Vietnamese potbellied pigs were also significantly older than other breeds, so it is unclear if this lymphocyte difference is an effect of age or a true breed difference. Mild differences were also noted based on breed for protein measurements, with lower values noted in American mini pigs. This group did not differ from other breeds based on sex or age, so it is possible that this may be a true breed difference, but there were only 13 American mini pigs in the study, so this would need to be evaluated in a larger number of individuals.

There are a few limitations of this study. According to ASVCP guidelines, the ideal group size for determining reference intervals is at least 120 individuals (*Friedrichs et al., 2012*). However, they acknowledge that in veterinary medicine there are issues that can make this goal difficult to achieve, such as availability of individuals or cost of sample collection and analytical procedures. For this reason, they also provide recommendations for determination of reference intervals in sample sizes between 40 and 120. Indeed, the goal of the present study was to include 120 pigs, but despite the prolonged course of the study, only 94 clinically healthy companion pigs were available. Additionally, some values were not available for all samples for various reasons, including reticulocytes, iron, triglycerides, and cholesterol. However, there were at least forty samples for most values, so the recommended methods for a sample size between 40 and 120 were used. The exception was reticulocytes, for which there were only 20 individuals. Due to the small sample size, a reference interval was not calculated for reticulocytes, rather descriptive statistics were provided.

Results at the extremities of distribution were retained in most cases to maximize sample size; however, a few outlying results were eliminated. These included the platelet measurements from a sample with numerous large platelet clumps, and high AST and CK values from two pigs suspected of having muscle injury during transport or sample collection. Additionally, two high GGT measurements were excluded from the two most lipemic samples in the study. The manufacturer of the GGT assay used in this study states that the degree of lipemia based on the lipemic index in those samples is not expected to

interfere with GGT measurement (Roche Diagnostics, Basel, Switzerland). This suggests that analytical interference from lipemia might not explain the high GGT results in these pigs, although the manufacturer's claim is based on experiments using an artificial lipid additive that may not correspond to endogenous porcine lipids. Both pigs were 9-month-old males that appeared clinically healthy, and neither had high total bilirubin measurements (0.1 mg/dl in both animals), arguing against cholestasis as a cause of increased GGT. A cause for the extreme GGT values in these individuals cannot be confirmed. Because this finding was only noted in the two samples with marked lipemia, an unverified lipemia interference is suspected; however, subclinical liver disease such as biliary hyperplasia or cholestasis cannot be ruled out.

Additional limitations include that these samples were measured on specific analyzers, so the resulting reference intervals may not be completely transferrable to other instruments. These results should be used as guidelines only if internal reference intervals are not available from the laboratory evaluating the samples. Also, the confidence intervals, particularly of upper reference limits, were often wider than ideal, indicating that mildly increased values may not always indicate pathology. Additionally, almost half of the reference group was made up of intact females, likely because many of the samples were collected prior to ovariohysterectomy. However, there were very few significant differences in results based on sex, so this might not have appreciably skewed reference intervals.

Finally, the majority of pigs were sedated or anesthetized prior to sample collection, with the most common drugs being xylazine, midazolam, ketamine, and isoflurane so we cannot exclude that some values may have been impacted. Specific effects of these drugs on hematology and biochemistry in companion pigs is uncertain, but there are reported effects in other species. The drugs used in our study have been associated with lower values for RBC concentration, hemoglobin, hematocrit, or packed cell volume in several species, possibly due to splenic sequestration and fluid shifting (*Custer et al., 1977*; *Dhumeaux et al., 2012*; *Kullmann et al., 2014*; *Marini et al., 1994*). Mild increases in CK were observed in dogs administered xylazine and ketamine (*Franco et al., 2009*), so it is possible that this might have affected the CK results from some of the pigs in our study. Hepatotoxicity and increased liver enzymes have been reported after repeated ketamine doses in people (*Cohen et al., 2018*), but ketamine was only administered once before blood collection in these pigs so this is considered unlikely. Ketamine is reported to cause increased plasma cortisol in calves, horses, and rabbits, so it is plausible that there may be cortisol-related effects on laboratory results, such as increases in neutrophils and glucose, and decreases in lymphocytes (*Plumb, 2024*). Lastly, isoflurane has been reported to cause mild increases in AST or GGT in some species (*Humann-Ziehank & Ganter, 2012*).

In contrast, unsedated pigs may become stressed during blood collection, with the resultant epinephrine response also impacting results, such as promoting increased concentrations of blood neutrophils, lymphocytes, and glucose. This highlights that animal handling and medications may impact hematology or biochemical results in various ways. It is therefore ideal that the blood collection procedure for the reference sample group is as similar as possible to that used for blood collection from the patient population in which the reference intervals will be used. At the UTVMC, clinical porcine patients are

commonly, but not always, sedated or anesthetized prior to blood collection, similar to the reference sample group in this study. Care should be taken in using these reference intervals in unsedated pigs, as alterations may result from the differences in patient preparation.

## CONCLUSIONS

This study provides hematology and biochemistry reference intervals for companion porcine patients, including a comparison of results based on age, sex, and breed. Some values were significantly different based on age, all of which have also been described in other species, but few values were significantly different based on sex or breed. To minimize bias introduced by differences in methodology, reference intervals are ideally generated *de novo* in the same laboratory in which patient samples are measured. However, because reference interval determination is a time-consuming and costly endeavor, these published values can be used cautiously to interpret data from companion pig patients when local reference intervals are not available.

## ACKNOWLEDGEMENTS

The authors acknowledge the excellent veterinary technical staff of the UTVMC farm animal hospital, especially Mary Passmore and Kaitlin Houser, for help with sample collection, and the outstanding medical laboratory scientists of the UTVMC clinical pathology laboratory for their expertise with sample processing and blood smear evaluation, including Chris Ramsey, Kylie Araghi, Kristen Brooks, Andrew Ferguson, Aimee Hebrard, and Angel Quiggle.

### Funding

The authors received no funding for this work.

### Competing Interests

The authors declare that they have no competing interests.

### Author Contributions

- Deanna M. W. Schaefer conceived and designed the experiments, analyzed the data, prepared figures and/or tables, authored or reviewed drafts of the article, and approved the final draft.
- Ricardo Videla conceived and designed the experiments, performed the experiments, authored or reviewed drafts of the article, and approved the final draft.
- Joe S. Smith performed the experiments, authored or reviewed drafts of the article, and approved the final draft.
- Pierre-Yves Mulon performed the experiments, authored or reviewed drafts of the article, and approved the final draft.
- Bente Flatland analyzed the data, authored or reviewed drafts of the article, and approved the final draft.

- Xiaojuan Zhu analyzed the data, prepared figures and/or tables, authored or reviewed drafts of the article, and approved the final draft.

## Animal Ethics

The following information was supplied relating to ethical approvals (*i.e.*, approving body and any reference numbers):

Blood samples were collected from 2019 to 2024 according to approved University of Tennessee IACUC protocols (UT IACUC protocol numbers 22980914 for 2019–2022 and 29470123 for 2023–2024).

## Data Availability

The hematology and biochemistry results for all study pigs are available in the Supplemental File.

## Supplemental Information

Supplemental information for this article can be found online at http://dx.doi.org/10.7717/peerj.18968#supplemental-information.

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
