# Peer review of "Hematology and clinical biochemistry reference intervals for companion pigs using the ADVIA 2120 and Cobas c501"

_PeerJ, doi:10.7717/peerj.18968_

## Round 0.1 · original submission · Minor Revisions

The reviewers have provided valuable perspectives and suggestions should be addressed. In particular reviewer 3 has several specific suggestions that will improve the clarity of the manuscript.

·

Basic reporting

BASIC REPORTING: Overall good
PROFESSIONAL ENGLISH LANGUAGE: Good
INTRO AND BACKFROUND FOR CONTEXT: Good
LITERATUIRE REVIEW RELEVANT AND WELL REF: Good
STANDARD CONFORMS: Yes
FIGURES RELEVANT, HIGH QUAL, LABELLED AN DESCRIBED: Yes
RAW DATA PROVIDED: Yes

Experimental design

EXPERIMENTAL (methodological) DESIGN: Good
WITHIN SCOPE OF THE JOURNAL: Yes
RESEARCH WELL DDEFINED, RELVANT, MEANINGFUL: Yes
INVESTIGATION HIGH TECH AND ETHICAL STANDARD: Yes
METHODS:
DESCRIBED SUFFICIENT DETAIL AND INFORMATION: Yes

Validity of the findings

VALIDITY OF THE FINDINGS: Good
MEANINGFUL REPLICATION ENCOURAGED: Contribute to a previously uninvvestigated population in the literature of companion pigs
UNDERLYING DATA PROVIDED: Yes
CONCLUSIONS WELL STATED, LINKED TO ORIGINAL RESERARCH AND LIMITED TO SUPPORTING RESULTS: Yes
VERTEBRATE ANIMAL USEAGE CHECKS: Not applicable
ETHICAL APPROVAL STATEMENT: Not applicable
WERE EXPERIMENTS NECESSARY AND ETHICAL? Not applicable
ANIMAL RESEARCH POLICIES: Not applicable

Additional comments

Excellent paper with good Discussion of various circumstances and physiological occurrences that may influence the results. The supplementary information is particularly good and I appreciated the comparison (Supplementary Table 4) with other authors’ results.

I strongly recommend this manuscript for publication and believe it will be of interest to the readers of PeerJ and to the veterinary laboratory community.

·

Basic reporting

The paper is well written and there is excellent attention to detail. The data provided, including the supplementary materials, are useful and thorough.

Experimental design

Could you describe whether the bloodsmears were checked for platelet clumps and if you opted to exclude those with platelet clumps?

Could you describe if the pigs included in the study were fasted? And if so, for how long?

Line 294: Did you look at plateletcrit data? Was it higher in the younger pigs as well?

Line 283: If MCV increases with age but the RBC count decreases, is there an effect on the hematocrit/PCV or does that stay constant?

Validity of the findings

Line 136-148: Could you clarify some aspects of these statistics? Is there an assumption that the SD of the data is consistent for this test, and if so, did you test for this? If this test compares differences between means, is it possible that it wouldn't be able to detect a difference in a situation where one group has a wider reference interval but both have the same means? Would it be better to look for differences in the lower (2.5th percentile) and upper values (97.5th percentile?) for the purpose of this study?

Additional comments

Line 370: you’re doing a great job summarizing the effect of sedation and anesthesia on the biochemistry profile. Please consider adding information about the changes seen on the CBC profile secondary to sedation/anesthesia as well.

Here is another reference interval paper in minipigs you may want to incorporate in the references (+/- introduction and discussion) and supplemental tables 3 and 4 of the paper:
Yeom SC, Cho SY, Park CG, Lee WJ. Analysis of reference interval and age-related changes in serum biochemistry and hematology in the specific pathogen free miniature pig. Lab Anim Res. 2012;28(4):245-253. doi:10.5625/lar.2012.28.4.245

Reviewer 3 ·

Basic reporting

Clear and unambiguous, professional English used throughout.
- The writing is professional, uses appropriate terminology and clearly explains each section.

Literature references, sufficient field background/context provided.
- Citations are appropriate to content and the introduction provides a logical argument conveying the merits of the study based on previous work. An additional citation to consider including is mentioned in the additional comments.

Professional article structure, figures, tables. Raw data shared.
- The article is written as instructed by the journal guidelines. Figures and tables are formatted correctly and easy to follow. The raw data file is present.

Self-contained with relevant results to hypotheses.
- Yes, the results and discussion relate directly to the hypotheses.

Experimental design

Original primary research within Aims and Scope of the journal.
- Yes, the manuscript is original research and falls within the scope of the journal.

Research question well defined, relevant & meaningful. It is stated how research fills an identified knowledge gap.
- Yes, the authors define the gap in the literature (reference intervals for companion pigs vs commercial or research pigs).

Rigorous investigation performed to a high technical & ethical standard.
- It is my opinion that the authors followed the ASVCP reference interval guidelines appropriately and thoroughly. Ethics protocols were in place and all samples were collected and analyzed according to common widely accepted methods.

Methods described with sufficient detail & information to replicate.
- Yes, the methods were described thoroughly.

Validity of the findings

Impact and novelty not assessed. Meaningful replication encouraged where rationale & benefit to literature is clearly stated.
- The rationale and benefit to the literature are clearly stated.

All underlying data have been provided; they are robust, statistically sound, & controlled.
- Yes, the raw data file has been provided. The methods used are those recommended by the ASVCP.

Conclusions are well stated, linked to original research question & limited to supporting results.
- Yes, I have no concerns about the conclusions drawn from the data.

Additional comments

Line 68 - The authors should consider adding Perri et.al, Hematology and biochemistry reference intervals for Ontario commercial nursing pigs close to the time of weaning, CVJ, 2017, 58:371-376. This paper is more recent than some of the other cited, assessed large numbers of pigs, data was generated using the ADVIA 2120 and Cobas c501analyzers and the authors mention following the ASVCP guidelines. This should be a relevant study to draw some comparisons with.
Line 89 – The potential effects of these drugs on hematological and biochemical variables is minimally described but a justification was provided in the discussion that most pigs will receive one or more of these drugs for blood collection in routine practice so I thought this was handled well.
Line 109 – I’m curious about the word ‘scientist’ used here – does this imply the WBC differentials were undertaken by individuals other than MLTs?
Line 137 – Just to make sure this is not an error, were there no pigs between 11months and 12 months of age?
Line 196 – I would encourage the authors to find a primary literature source to back up this statement rather than cite a website.
Line 277 - I would encourage the authors to find a primary literature source to back up this statement rather than cite a textbook. This may be acceptable if a primary source can’t be found.
Line 279 – Does this statement make sense with the lifespan of porcine erythrocytes?
Line 282 – How many pigs of the 42 animals with an iron measurement were in the younger group? This would add some context to the findings.
Line 287 – I question whether it’s appropriate to call it erythrocytosis when the values derive from healthy animals and are forming part of the reference values. Using this term implies an increase beyond what is expected and the values here are instead typical for these animals (so you argue).
Line 292 – Similar comment about the textbook citation.
Line 304 – It would be helpful to speculate on why the protein concentration is lower.
Figures 2&3 – are the higher lymphocyte (8-10x10^9/L) and neutrophil (>10x10^9/L) counts the same animals but both automated and manual methods?

---

## Round 0.2 · accepted · Accept

Well done - it was a pleasure handling your manuscript.

·

Basic reporting

In my opinion, the suggestions from all reviewers were addressed well. Thank you for making these changes!

Experimental design

Adequate

Validity of the findings

Adequate

Reviewer 3 ·

Basic reporting

I am satisfied with the changes the authors have made to this section.

Experimental design

I am satisfied with the changes the authors have made to this section.

Validity of the findings

I am satisfied with the changes the authors have made to this section.

Additional comments

N/A